# The Potential Role of Human Milk Oligosaccharides in Irritable Bowel Syndrome

**DOI:** 10.3390/microorganisms10122338

**Published:** 2022-11-25

**Authors:** Patricia Sanz Morales, Anisha Wijeyesekera, Margaret Denise Robertson, Peter P. J. Jackson, Glenn R. Gibson

**Affiliations:** 1Department of Food and Nutritional Sciences, University of Reading, Whiteknights, Reading RG6 6AH, UK; 2Department of Nutritional Sciences, Faculty of Health & Medical Sciences, University of Surrey, Guildford GU2 7XH, UK

**Keywords:** human milk oligosaccharide, irritable bowel syndrome, gut microbiota, 2′fucosyllactose, *Bifidobacterium*

## Abstract

Irritable Bowel Syndrome (IBS) is the most common gastrointestinal (GI) disorder in Western populations and therefore a major public health/economic concern. However, despite extensive research, psychological and physiological factors that contribute to the aetiology of IBS remain poorly understood. Consequently, clinical management of IBS is reduced to symptom management through various suboptimal options. Recent evidence has suggested human milk oligosaccharides (HMOs) as a potential therapeutic option for IBS. Here, we review literature concerning the role of HMOs in IBS, including data from intervention and in vitro trials. HMO supplementation shows promising results in altering the gut microbiota and improving IBS symptoms, for instance by stimulating bifidobacteria. Further research in adults is required into HMO mechanisms, to confirm the preliminary results available to date and recommendations of HMO use in IBS.

## 1. Introduction

Irritable bowel syndrome (IBS) is a highly prevalent gastrointestinal (GI) disorder with significant negative impact on quality of life of patients and high healthcare costs [1]. Although prognosis of IBS is benign, it is a disorder that poses a considerable burden on the individual sufferer and society. Patients typically present with chronic abdominal pain and an altered bowel habit, frequently accompanied by bloating and distension. Often, IBS will afflict sufferers for life, with flares of activity followed by periods of remission [2]. Incidence commonly peaks in the third and fourth decades of life [3].

IBS is suggested to be a disorder of gut-brain interaction, and alterations of the microbiota-host interactions at the mucosal border may cause symptoms such as those previously mentioned. Therefore, microbiota-targeted interventions may benefit some people with IBS by beneficially modulating the gut microbiome [4]. Several studies have confirmed that prebiotics, such as galactooligosaccharides (GOS), are able to successfully stimulate gut bifidobacteria and alleviate symptoms in IBS [5,6]. Prebiotics are defined as “a substrate that is selectively utilised by host microorganisms conferring a health benefit” [7]. These studies suggest that prebiotics may have potential as therapeutic agents in IBS.

Breastmilk is known to play a crucial role in the development of infants, providing key nutrients and immunological compounds important for initial protection against pathogens [8]. Among these compounds, human milk oligosaccharides (HMOs) represent the third most important component of breastmilk after lipids and lactose [9]. HMOs have also been investigated for potential health benefits in adults, including their potential role as prebiotics for improved gut microbiota modulation [10].

## 2. The Burden of IBS

Definition of IBS have evolved over time, and a single test with reliable sensitivity and specificity is yet to be established. Manning criteria were the first diagnostic approach to be used globally, introduced in 1978 [11]. They were based on symptoms thought to occur more frequently in people with IBS compared to organic disease. However, sample sizes used in the original publication was small (32 with IBS vs. 33 with organic disorder), and the sensitivity ranged from 63–90% [11]. Organic diseases are those in which there is a structural alteration to an organ or part. Organic gastrointestinal diseases may mimic symptoms of IBS, including colorectal cancer and coeliac disease [12].

In the United Kingdom, the National Institute for Health and Research Excellence (NICE) defines IBS as “abdominal pain or discomfort, in association with altered bowel habit for at least 6 months, in the absence of alarm symptoms or signs” [13]. IBS subtypes are established according to stool consistency, for example using the Bristol Stool Chart [14]. IBS may be classified into IBS-constipation (IBS-C), IBS-diarrhoea (IBS-D), IBS-mixed (IBS-M) and IBS-unspecified (IBS-U) [15]. Current NICE guidelines for management of IBS recommend diagnosis using clinical grounds alone, without a need for invasive investigations, unless warnings such as weight loss or rectal bleeding are present [13]. British Society of Gastroenterology guidelines on the management of IBS suggest that the NICE definition of IBS may be more suitable in a primary care setting, whereas Rome criteria may be better suited in secondary care [16].

To aid diagnosis of IBS, Rome stool criteria have been employed using predominant stool patterns (Table 1 and Figure 1). These criteria were developed by the Rome Foundation and have been updated and fine-tuned over the years to better suit clinical practice. The latest update being the Rome IV criteria (May 2016) [1].

Diagnostic criteria *.

Recurrent abdominal pain on average at least 1 day/week in the last 3 months, associated with two or more of the following criteria:Related to defaecationAssociated with a change in frequency of stoolAssociated with a change in form (appearance) of stool

* Criteria fulfilled for the last 3 months with symptom onset at least 6 months prior to diagnosis. Adapted from [1].

Despite these criteria, the global prevalence of IBS in adults remains elusive due to the heterogeneity of reporting [18]. A meta-analysis by Oka et al. [19] using the Rome III criteria, concluded that global distribution of IBS was 27.8% IBS-D, 20.0% IBS-C and 33.8% IBS-M. When Rome IV criteria were used, the distribution was: 31.5% IBS-D, 29.3% IBS-C and 26.4% IBS-M. These differences may arise from the use of different diagnostic techniques, amongst other limitations such as country prevalence variability, which might skew results. Overall global prevalence of IBS has been found to be 9.2% when Rome III criteria were used, and 3.8% with Rome IV criteria. Proportions of persons with IBS are lower when the Rome IV criteria are used, compared with the Rome III criteria [20]. Rome IV criteria are more restrictive and hence may be less suitable for population-based epidemiological studies. They were modified based on a large population study with the goal of increasing sensitivity and specificity of the criteria [17].

IBS patients live with the burden of relapsing and remitting symptoms such as abdominal pain, mental health issues such as anxiety and depression, bloating, diarrhoea, and constipation [21]. A study of 300 IBS patients (IBS-M, 45.1%; IBS-D, 39.2%; IBS-C, 15.7%) by Ringel et al. [22] concluded that the most bothersome symptoms reported were abdominal pain or discomfort, cramping and bloating. Most of these symptoms are not required for diagnosis, as per Rome IV criteria but are consistently reported by IBS patients [1].

There is pressing need for refining IBS pathophysiology, although symptoms vary considerably over time both within and between patients, posing a challenge. Moreover, symptoms employed to determine IBS are non-specific and can originate from other disorders such as inflammatory bowel disease or coeliac disease, for instance [23].

### 2.1. IBS Pathophysiology

Pathophysiology of IBS has traditionally focused on the importance of visceral hypersensitivity and development of pain and discomfort [24]. On the other hand, impacts of gut dysmotility has been largely integrated within a more holistic concept of IBS aetiology, alongside dysregulated gut-brain axes [25]. IBS is a syndrome resulting from the disturbed interaction between psychosocial, physiological, behavioural and environmental factors [26].

It has been long suspected that stress is related to IBS symptoms [27]. Evidence now indicates that chronic, sustained stress is related to IBS symptomatology, particularly flare-ups, and that IBS is a disorder involving the gut-brain axis [28]. The intestine is considered to be a ‘second brain’, being part of the enteric nervous system, giving rise to a complex interconnection widely known as the gut-brain axis [29].

People with IBS frequently report that symptoms are also generated or exacerbated by certain meals, and about 50% with IBS-D complain of postprandial diarrhoea [30], potentially due to a lessened ability to accommodate post-prandial inflow in the ascending colon. Evidence suggests that people with IBS-D have faster orocecal transit times and smaller terminal ileum diameter compared to healthy controls [31,32]. This may account for increased postprandial pain observed in IBS-D cases.

Differences in clinical features between the various IBS subtypes have made it particularly difficult to develop interventions and propose new treatments [33]. Moreover, analyses of current treatment option suitability and effectiveness is hindered by varying IBS symptomology. Some patients have relatively mild symptoms and do well, whereas others experience severe symptoms that are deleterious to their quality of life [34]. This therefore suggests that IBS subtypes should be clinically managed separately, to allow for a more personalised approach [35]. A recent study in over 4000 IBS sufferers divided patients into seven characterised clusters [36]. The clusters identified varying degrees of GI symptoms, comorbidities, dietary, and lifestyle factors. Future intervention trials should assess whether these unique clusters could be used to direct clinical trials and individualise patient management.

### 2.2. Gut Microbiome in IBS

The human GI tract microbiota plays a pivotal role in human health. Establishing relevant connections between the host and presence or abundance of specific microbial communities is complicated and requires high-throughput technologies [37,38]. Through this, changes in gut microbiota, increased mucosal permeability, and low-grade inflammation have all been associated with the pathophysiology of IBS [39]. There is evidence to suggest that underlying gut dysbiosis may lead to activation of the gut immune system with downstream effects on a variety of other factors of potential relevance to the pathophysiology of IBS [40,41].

It has been proposed that the microbiota of IBS patients differs to that of healthy controls [42]. An in vitro global and deep molecular analysis of IBS faecal samples indicated a 2-fold increased ratio of Firmicutes to Bacteroidetes (*p* = 0.0002) in 62 Rome II-defined IBS patients compared to 46 controls [38]. Other significant signatures of IBS included a 10% decreased level of Actinobacteria (*p* = 0.034), with a 1.5-fold reduction of *Bifidobacterium* (*p* < 0.05) and a >5% increased level of Firmicutes (*p* < 0.0001). The study had several strengths, including design and IBS subtype representability (25 IBS-D, 18 IBS-C, 19 IBS-M). A redundancy analysis plot clearly illustrated differences in healthy microbiota vs. IBS microbiota. However, the small sample size may have compromised power and IBS female:male ratio was very high (57:5) [38]. Notwithstanding, these results have since been corroborated by other studies [43,44].

A study by Dunlop et al. [41] indicated that small intestinal permeability is frequently abnormal in IBS-D patients. Although the power of the study was quite low, IBS subtypes were clearly defined, and significant results were found. These findings, coupled with associated low-grade inflammation, could explain why IBS patients have a different gut microbiome compared to healthy controls [45]. It is feasible that microbial metabolites of the disturbed gut microbiome can trigger responses that cause IBS pathology [46]. Nevertheless, there has been no definite microbiome signature found to date for IBS [47]. This may be partly due to the heterogeneity of studies available, but also due to the heterogeneity of healthy gut microbiomes, which makes it difficult to draw specific conclusions [48].

## 3. Current Treatment Options

Existing treatment for IBS remains conflicting and suboptimal. With no wholly effective cure, IBS treatment is dependent on addressing the patients’ most troublesome symptoms [49]. Current approaches rely on explanation of the diagnosis, lifestyle/dietary advice and medication to manage main symptoms [50].

Pharmacological treatment modalities for IBS target GI receptors and ion channels, peripheral opioid receptor, gut serotonin receptors, and gut microbiota [51] (Figure 2). Selective serotonin reuptake inhibitors and tricyclic antidepressants are effective in numerous IBS cases, suggesting the implication and dysfunction of the central or peripheral serotonergic system in IBS pathophysiology [52,53]. Psychological therapies also appear to be effective treatments for IBS, although there are limitations in the quality of evidence, and treatment effects may be overestimated as a result [54].

Antibiotic treatment has been deemed safe for IBS-D, with mixed response regarding bloating, abdominal pain and stool consistency [55]. The Food and Drug Administration (FDA) has approved rifaximin (Xifaxan ^®^) as the only antibiotic for IBS treatment to date with a recommended 2-week dosing period [56,57]. Alongside rifaximin, the FDA has approved alosetron and eluxadoline for IBS-D treatment [54]. Serotonin (5-hydroxytryptamine, 5-HT) receptor subtypes are believed to be involved in the pathophysiology of IBS [57]. Briefly, alosetrone is a 5-hydroxytryptamine 3 (5-HT3) receptor antagonist. 5-HT3 receptor stimulation is associated with increased GI motility and secretion. Eluxadoline is a μ and κ-opioid receptor agonist and δ-opioid receptor antagonist that acts locally at the level of the gut mucosa and enteric nervous system to reduce both colonic secretions and GI transit [58].

Non-pharmacological IBS treatment options such peppermint oil (PO) is often recommended [59,60]. The likely mode of action of PO is as anti-spasmodic, although there is evidence that PO has a range of other effects including anti-nociception, anti-inflammatory, and carminative effects [61,62]. In a randomised controlled trial (RCT), PO significantly reduced IBS symptom scores by day 28, compared to placebo [63]. However, this therapy option was indicated only for non-constipated IBS patients, excluding IBS-C. Moreover, the subtypes of IBS were not classified or represented appropriately, instead naming volunteers “subjects with moderate to severe non-constipated IBS” [63].

Alternative treatments for IBS-C include laxatives, which can be categorised depending on their mode of action into osmotic, bulking and stimulant [64]. The most recognised of all bulking laxatives being psyllium [65], while osmotic laxatives encompasses polyethylene glycol, lactulose and Movicol [66]. Stimulant laxatives including Biscodyl and senna [67].

Other non-pharmacological management options such as dietary modification can be a primary management strategy for IBS, particularly use of dietary fibre [61,68]. Many IBS-D patients suffer from postprandial diarrhoea, and hence associate certain foods with flare-ups [31,32]. The low fermentable oligo, di, monosaccharides and polyols (FODMAP) diet was devised to relieve symptoms in IBS patients. A research group at Monash University in Australia undertook the first investigations to prove whether a low FODMAP diet could improve symptom control in IBS patients and established a mechanism by which this diet exerted its effect [69,70,71]. A retrospective study by Staudacher et al. [72] comparing standard dietary advice to the low FODPMAP diet in 82 IBS patients found a significant improvement in overall symptom response in the low FODMAP diet group (*p* < 0.001). However, in this study the use of other medication was not controlled for, and IBS subtypes were not distinguished. Moreover, participants completed the questionnaire at the time of review consultation, which may have introduced bias [72].

In a crossover study conducted in Australia, 30 IBS patients and 8 healthy controls were randomly assigned to either a low FODMAP diet or a typical Australian diet (a healthy, balanced diet recommended by the Australian government [73]) for 21 days. A washout period of at least 21 days between alternate diets was included. Results supported the use of low FODMAP diet as a first-line therapy for IBS, as subjects had lower overall GI symptom scores (22.8; 95% confidence interval (CI), 16.7–28.8 mm) while on a diet low in FODMAPs, compared with the Australian diet (44.9; 95% CI, 36.6–53.1 mm; *p* < 0.001) and the subjects’ habitual diet [74]. With this type of randomised, double-blinded study design, a major limitation is the fact that subjects easily identify the diet they were on, and it is therefore hard to exclude a placebo effect. Another limitation was the small size of the study.

FODMAPs do not necessarily cause the underlying disorder but represent an opportunity for reducing IBS symptoms. Nevertheless, the use of restrictive diets may not be ideal and may lead to potential nutrient deficiencies if prolonged. Foods that provide a key source of micronutrients in habitual diets such as legumes and dairy products may be removed, creating a void. For instance, Staudacher et al. [72] reported a diminished calcium consumption in the 4-week restrictive diet trial. Moreover, there is a lack of studies investigating the re-introduction of FODMAP foods into the diet of IBS patients.

Reintroduction of FODMAP foods is a critical part of the treatment plan, as there are long-term negative consequences of staying on a low FODMAP diet. For instance, the gut microbiome is severely affected, since there is a reduction in prebiotic/fibre consumption and studies have shown a decrease in *Bifidobacterium* [74], although this microbiota change has only been studied short-term and it is unknown whether this change would persist long-term [75]. Hence, there is a need for interventions that assess long-term effects of low FODMAP diets on the gut microbiome and associated health consequences.

Moreover, according to a study by Maagaard et al. [76], many IBS sufferers who try the low FODMAP diet have low adherence due to increased difficulty in continuing the diet, high cost, and bland taste. Higher costs of low FODMAP diets compared to habitual diets was confirmed by an additional study [77].

### Prebiotic Supplementation

There is increasing evidence for a role of the gut microbiota in IBS pathogenesis. Variations in faecal microbiota, the use of probiotics/prebiotics, and recognition of an upregulated host immune system response suggest that an interaction between the host and GI microbiota may be important in the pathogenesis of IBS [78]. For instance, a RCT in 33 children with Rome III-defined IBS reported that baseline gut microbiome characteristics identified IBS patients who were more likely to respond to the low-FODMAP diet and effectively decrease abdominal pain frequency [79].

Oligosaccharides such as fructooligosaccharides (FOS) and GOS resist hydrolysis by saliva and intestinal digestive enzymes and are considered prebiotics [5,80]. Configuration of glycosidic bonds and lack of brush border glycosidases allows these polymers to reach the colon and be fermented by anaerobic bacteria. For instance, once in the colon, FOS are rapidly broken down to short-chain fatty acids (SCFAs) (acetate and lactate), mostly by bifidobacteria [81,82]. Enzymes such as β-fructofuranosidase found in bifidobacteria can break down FOS [83,84].

Since gut microbiome-mucosal interactions may be involved in the pathogenesis of IBS, FOS and GOS have been trialled as potential therapy options, with variable results. A systematic review by Ford et al. in 2018 identified three RCTs, two using FOS and one using GOS as potential prebiotics for IBS treatment [85]. A RCT using Manning criteria recruited 98 IBS patients and randomised them to receive either 20 g of FOS powder or placebo, for 12 weeks [86]. Data from 96 patients was analysed (16 men, 80 women). Upon completion of the study, there were no differences between both groups. This multicentre, double-blind trial had low risk of bias and adverse events were similar in each arm [86].

A second RCT recruited 79 patients with Rome III-defined IBS and used short-chain FOS (scFOS) vs. placebo for 4 weeks [87]. There were no differences in global symptom scores between both treatment arms, likely due to the low power of the study. However, it was found that the effect of scFOS on rectal sensitivity was more pronounced in constipation-predominant-IBS patients (*p* = 0.051 vs. placebo). Other limitations, such as the unreported method used to conceal treatment allocation, may have introduced bias [87].

60 patients with Rome II-defined IBS were included in another well-designed crossover trial [5]. Results indicated that GOS significantly increased *Bifidobacterium* compared to placebo (*p* < 0.05) and in turn ameliorated global IBS symptom scores, however no effect on abdominal pain was seen. This study was on unclear risk of bias, as the method of randomisation was stated, but not the method of concealment of allocation. The trial was single-blinded as only patients were blinded to treatment allocation. Again, adverse events were similar in all arms [5].

## 4. Human Milk Oligosaccharides

HMOs are the third most abundant component of human milk, following lactose and lipids [88]. They are minimally digested in the GI tract and reach the colon intact, being fermented by the gut microbiota [89]. Discovered in the early 1930s, HMOs were originally thought to be the “bifidus factor” of human milk [90]. Since then, HMOs have been suggested to offer numerous additional benefits to the developing neonate. We now know that oligosaccharides represent one of the most relevant components of human milk, with their effects extending past infancy, into adulthood [91].

HMOs are a family of structurally diverse glycans unique to human milk, composed of the five monosaccharides: glucose (Glc), galactose (Gal), N-acetylglucosamine (GlcNAc), fucose (Fuc) and sialic acid (Sia), with N-acetylneuraminic acid (Neu5Ac) as the predominant, if not only, form of Sia [92]. All HMOs contain a lactose molecule at the reducing end and may be sialylated or fucosylated (Figure 3).

The most extensively researched HMOs to date include 2′-fucosyllactose (2′FL), 3-fucosyllactose (3′FL), 3′-sialyllactose (3′SL), 6′-sialyllactose (6′SL), Lacto-N-*neo*tetraose (LNnT) and Lacto-N-tetraose (LNT) (Figure 4) [93,94]. Some research has been carried out on bovine milk oligosaccharides (BMOs), although these are 20–100 times less concentrated than in human milk [95,96]. BMOs are structurally related to HMOs, however HMOs are produced more consistently during lactation process whereas BMOs decrease considerably even during the first few days of lactation [97,98].

Over 200 different HMOs have been identified to date, although not every female would synthesise the same set of HMOs [99]. All HMOs are synthesised in the mammary gland, although synthetic production of HMO may be performed by (chemo-)enzymatic syntheses or by whole-cell biotransformation with recombinant bacterial cells [100].

The HMO composition of breastmilk (both the amount and presence of specific structures) differs between mothers, depending on their genetic profiles. Fucosylated HMO profile is determined by the secretor status and Lewis blood type of each mother [101], although recent findings suggest that other factors play a role in determining the compositional nature of breastmilk [102]. For instance, α1-2-fucosyltransferase (FUT2 encoded by the Se gene) and α1-3/4-fucosyltransferase (FUT3 encoded by the Le gene) determine whether 2′FL is produced or not (Table 2).

Only a subset of females (“secretor” mothers) will synthesise 2′FL [103]. However, substantial differences in HMO profile can still be observed even after considering Lewis antigen system and secretor status. This may be in part due to competition for the same substrates by FUT2 and FUT3 [104].

### 4.1. HMOs in the Gut

Since the majority of HMOs reach the large intestine intact, they may serve as fermentable substrates for bacterial metabolism [105]. HMOs may be metabolised via intracellular as well as extracellular pathways [106,107], which have implications regarding utilisation of HMO breakdown products by other bacteria. Bacteria capable of utilising HMOs in the gut include bifidobacterial strains (frequently found as the most abundant colonisers in the gut of breastfed infants), such as *B. breve*, *B. bifidum*, *B. longum* subsp. *longum* (*B. longum*), *B. longum* subsp. *infantis* (*B. infantis*), and, in some cases, *B. pseudocatenulatum* [108,109,110]. Some *Roseburia* and *Eubacterium* strains also contain loci for HMO metabolism [111]. The Gram-negative anaerobe *Akkermansia muciniphila* is specialised in mucus degradation. It is also able to metabolise HMOs, due to their structural resemblance to mucus glycans [112] (studies summarised in Table 3).

It has been proposed that HMOs provide protection against infection by acting as decoys for pathogen binding [113]. For instance, a study by Ruiz-Palacios et al., showed that 2′FL inhibited *Campylobacter jejuni* colonisation of mice in vivo and human intestinal mucosa ex vivo [114]. *C. jejuni* binds to the intestinal H2 antigen, which contains a terminal α1-2-linked fucose residue. This study highlights a plausible mechanism behind the reduced incidence of infection observed among infants receiving milk from secretor mothers whereby the presence of α1-2-linked fucose on HMOs can act as pathogen decoys [115].

**Table 3 microorganisms-10-02338-t003:** Potential HMO utilisers in the gut.

Author ^1^	Type of Study	Main Findings	Reference
Gotoh et al., 2018	HMO-supplemented infant faecal media	*B. bifidum* grew in presence of 2′FL, 3′FL, LNT and LNnT.	[110]
Garrido et al., 2015	HMO-supplemented modified MRS media	*B. infantis* grew in presence of 2′FL, 3′FL, 6′SL and LNT.	[116]
Ryan et al., 2021	Batch culture fermentation and pilot clinical trial	*Bifidobacterium* grew in presence of 2′FL.	[117]
Lawson et al., 2020	HMO-supplemented media	*B. pseudocatenulatum* grew in presence of 2′FL and LNnT.	[118]
James et al., 2016	HMO-supplemented media	*B. breve* grew in presence of LNT and LNnT.	[119]
Garrido et al., 2016	HMO-supplemented media	*B. longum* grew in presence of LNT.	[107]
Kostopoulus et al., 2020	HMO-supplemented basal media	*Akkermansia muciniphila* grew in presence of 2′FL, 3′SL, LNnT, LNFPI, LNFPIII, 6′SL, LNT, LNFPV and DFL.	[112]

^1^ Main author and year study published. Key: 2′FL (2-fucosyllactose); 3′FL (3-fucosyllactose); LNT (lacto-N-tetraose); LNnT (lacto-N-neotetraose); 6′SL (6-sialyllactose); 3′SL (3-sialyllactose); LNFPI (lacto-N-fucopentaose I), LNFPIII (lacto-N-fucopentaose III); LNFPV (lacto-N-fucopentaose V); DFL (difucosyllactose).

A recent review by Jackson et al. [106] inferred that the metabolic fate of HMOs is more complex than previously thought. For instance, several microorganisms found within the gut microbiome, such as *B. adolescentis* and *B. animalis* do not grow well on HMOs individually in vitro [118,120]. Yet, these microorganisms may still be able to proliferate via cross-feeding on breakdown products of extracellular metabolism from HMO-degrading bacteria.

Since >90% of HMOs reach the large intestine intact, they may act as prebiotics. Recent findings have suggested HMOs as potential therapeutic agents for adults with gut conditions, such as IBS [6,121].

A study by Elison et al. [122] indicated that doses as high as 20 g/day of 2′FL and LNnT were safe and well tolerated in a group of 100 healthy adults. The supplementation period of 2 weeks demonstrated that HMOs modulate adult gut microbiota, with primary impact being substantial increases in Actinobacteria and *Bifidobacterium* spp. and decreases in Firmicutes and Proteobacteria. However, the study did not control for dietary composition which may be a confounding factor. Adverse events were seen in 44 subjects. Reports were mostly from the highest dose group (20 g) and related mainly to GI symptoms, particularly flatulence, stomach pain, diarrhoea/loose stools, and rumbling, and were characterised as mild. The intervention had a minor impact on stool frequency compared to placebo (an extra 0.3 bowel movements/day) which was not considered to be clinically important. It could be speculated that clinical relevancy may be found in higher-powered studies. However, this study was conducted in a healthy population and the safety of such a high dosage of HMOs to treat dysbiosis in GI conditions requires further study.

### 4.2. Potential Role of HMOs in IBS

IBS is characterised by deviations in gut microbiota [123]. Particularly, lower bifidobacterial counts have been noted in both faecal and mucosally associated microbiota [44]. Since HMO studies in healthy adults showed promising results [122], their role in shifting IBS microbiota towards a ‘healthy’ form has recently been explored.

Certain bifidobacteria possess genes related to degradation of HMOs, including sialidases, fucosidases, hexosaminidases, α-N-acetylgalactosaminidases, α-mannosidases, and lacto-N-biosidases [124]. For instance, *B. longum* subspp. *infantis* is capable of intracellular degradation of a wide variety of HMOs, making it a key facilitator of cross-feeding via acetate production [125,126]. This cross-feeding in turn stimulates changes in the wider gut microbiota, including increases in butyrate and propionate producers, such as *Faecalibacterium* and *Roseburia* [127]. Increases in these beneficial microbes may consequently promote gut barrier function [128,129]. Furthermore, metabolites from extracellular degradation of HMOs by other bifidobacteria, such as lactose, may promote growth of *Lactobacillus* [130].

Only two human trials investigating the role of HMOs in adults with IBS have been conducted to date [6,121]. Neither included healthy participants. Therefore, they did not evaluate if the microbiota profile of IBS patients responding to the HMO mix changed toward a profile resembling the healthy population.

The first trial by Iribarren et al. [121] was a phase II, parallel, RCT in 58 IBS volunteers. Intervention was placebo (glucose), or 5 g or 10 g 2′FL/LNnT for 4 weeks followed by a 4-week follow-up period. The primary endpoint was to determine the dose of 2′FL/LNnT that increases *Bifidobacterium* spp. abundance without aggravating GI symptoms, measured by gastrointestinal symptom rating scale-IBS (GSRS-IBS). As secondary efficacy endpoints, IBS severity, measured by IBS-symptom severity score (IBS-SSS), bowel habits (stool consistency), and anxiety and depression were assessed. Moreover, the effect on faecal microbiota and the proportion of responders, defined as a patient with a bifidobacteria abundance increased ≥50% at the end of the intervention period, relative to baseline were explored. Relevant results indicated that 10 g of 2′FL/LNnT increased *Bifidobacterium* spp. Although, no differences in overall GI symptom severity, anxiety or depression were found. Size of the study population was small, which likely also influenced the fit and predictive ability of microbiota profile analysis models.

The second trial by Palsson et al. [6] was open-label and included 245 IBS participants from 17 sites across USA. It was a prospective, single-arm clinical trial. A trial duration of 12 weeks was selected to provide sufficient time for stabilisation of symptoms and decrease of the placebo effect, while retaining the exploratory nature of the trial. Participants were administrated 5 g 2′FL/LNnT in a 4:1 mix daily orally. Bowel habits, IBS symptoms, and quality of life (QoL) were assessed at baseline and every 4 weeks. Patients had significant improvement from baseline to 12 weeks in total % of bowel movements with abnormal stool consistency, QoL and IBS-SSS.

Relatively short intervention periods did not allow the study of long-term effects, and this should be considered for future studies. Longer treatment periods may lead to an effect on both *Bifidobacterium* spp. and clinically relevant endpoints.

## 5. Discussion

Most HMO studies originally sought to understand the effects of HMOs on neonatal health, and to date research on the impact of HMO treatment in adults is more sparse. As neonatal growth progresses, new foods are introduced, replacing breastmilk and infant formulas. This diet change is accompanied by microbiome development, establishing an adult gut microbiota at approximately 2 years of age [131,132].

HMOs serve as prebiotic components supplying metabolic substrates necessary for beneficial bacteria to thrive (Table 3) [133]. Additionally, studies suggest that their prebiotic potential is not the only mechanisms via which HMOs exert positive health benefits [114], and therefore may be more suitable for IBS compared to FOS [87]. HMOs also function as soluble decoy receptors, preventing pathogen attachment to mucosal surfaces and improving gut barrier function [134,135]. There is a lack of clinical trials and in vitro work to confirm these effects of HMOs in adults. It is also hard to pinpoint which of the various mechanisms of action of HMOs would produce beneficial results in IBS studies, or whether they could be used as prevention strategies for post-infectious IBS. In addition, HMOs have mostly been studied in isolation, yet they exist as a diverse pool and interact in the mammary gland, infant and adult gut. Novel combinations of methodologies will be essential in future studies to unravel the multifaceted interactions, specific or otherwise, between HMOs and the gut microbiota [108].

Studies summarised in Table 3 emphasise that, for instance, not all *Bifidobacterium* strains possess the same glycosidases to degrade whole HMOs. Existing evidence is limited to pre-clinical models or infant-based studies yet suggests HMO metabolism to be strain specific. Moreover, in vitro studies fail to capture cross-feeding effects which may occur in vivo [106]. HMO degradants may be symbiotically shared among different bacterial species such as *Bifidobacterium* in the gut community. Additionally, other bacteria, such as the butyrate-producer *Blautia*, will utilise breakdown products from HMO degradation [136]. Therefore, the presence, abundance, and proportion of HMO utilisers in the gut may lead to mixed results seen in trials [137].

IBS has been associated with disruptions to the intestinal microbiota, however, studies have had limited power, coverage, and depth of analysis [42]. Microbiota composition analysis has confirmed that there are significant differences between the gut microbiota of healthy adults and patients with IBS [38]. However, the extent to which this information may help to better understand IBS, restore abnormal intestinal permeability, and develop new treatments is unclear yet promising [41]. Current data available from a limited number of patients, stating for instance *Bacteroides:Firmicutes* ratio, does not reveal marked and reproducible IBS-related deviations of whole phylogenetic or functional microbial groups, but rather supports the concept that IBS patients have alterations in the proportions of commensals with interrelated changes in the metabolic output and overall microbial ecology. A lack of apparent similarities in the taxonomy of microbiota in IBS patients may partially arise from the fact that the applied molecular methods, the nature, and location of IBS subjects, and statistical power of the studies have varied considerably [138].

## 6. Conclusions

Deviations in gut microbiota are associated with disorder [123,139]. The role of HMOs in protecting against disease through the modulation of the microbiome is clear and requires further examination, particularly in gut conditions such as IBS [140]. Moreover, the extent to which dysbiosis seen in IBS patients influences pathophysiology requires further exploration. Small sample sizes and variations in methodology of current available studies prevents firm conclusions. Furthermore, heterogeneity of ‘healthy’ microbiomes creates a problematic comparison with potential signatures for IBS [48].

To date, two published trials have looked at the effects of HMOs in people with IBS [6,121]. The prebiotic and other health benefits of HMOs may improve symptoms in IBS, a syndrome that is still managed poorly in clinical practice. However, existing evidence is limited to a 2′FL/LNnT HMO mix and poor evidence supports the use of a 4:1 ratio in adults. Future studies with other combinations of HMOs are required to demonstrate their potential health benefits and mechanisms in adults with IBS.

## Figures and Tables

**Figure 1 microorganisms-10-02338-f001:**
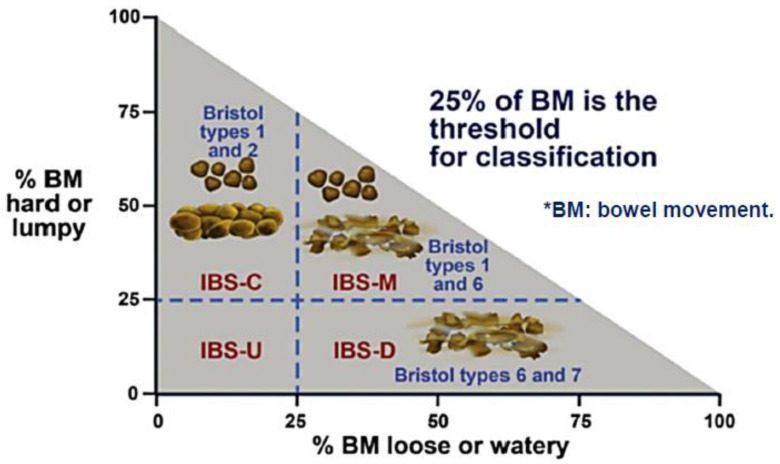
Rome IV criteria for IBS (May 2016).

**Figure 2 microorganisms-10-02338-f002:**
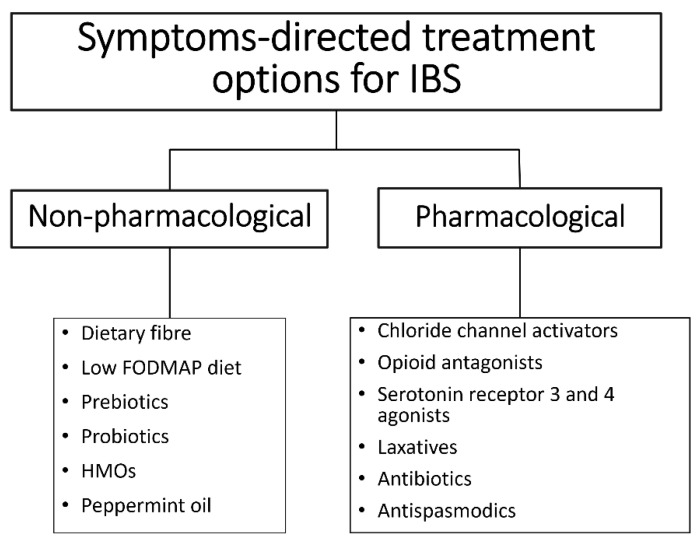
Treatment options for IBS are based on symptoms and may be pharmacological or non-pharmacological.

**Figure 3 microorganisms-10-02338-f003:**
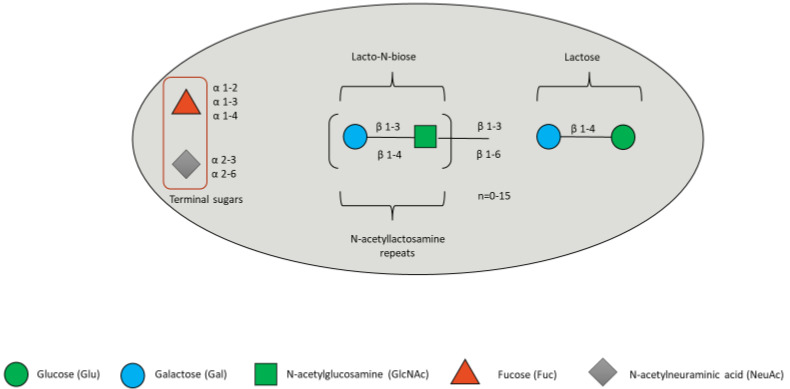
HMO composition blueprint. HMOs can contain 5 different monosaccharides in different number and linkages, namely glucose (green circle), galactose (blue circle), N-acetlylactosamine (green square), fucose (red triangle), and sialic acid (grey diamond).

**Figure 4 microorganisms-10-02338-f004:**
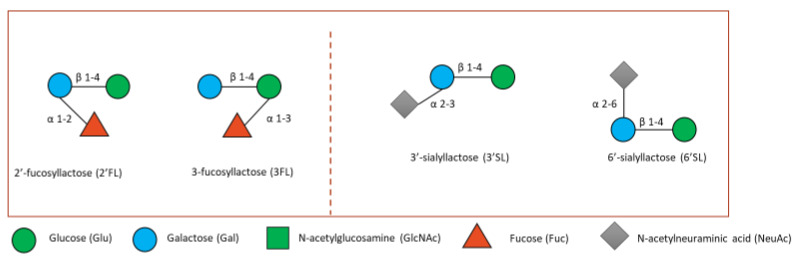
Small fucosylated (**left**) and sialylated HMOs (**right**). Lactose can be fucosylated or sialylated to generate the small HMOs 2′FL and 3′FL or 3′SL and 6′-SL, respectively shown here.

**Table 1 microorganisms-10-02338-t001:** Rome IV diagnostic criteria for IBS subtypes [17].

IBS Subtype ^1^	Diagnostic Criteria
IBS with predominant constipation (IBS-C)	>25% of bowel movements with Bristol stool types 1 or 2 and <25% of bowel movements with Bristol stool types 6 or 7.
IBS with predominant diarrhoea (IBS-D)	>25% of bowel movements with Bristol stool types 6 or 7 and <25% of bowel movements with Bristol stool types 1 or 2.
IBS with mixed bowel habits (IBS-M)	>25% of bowel movements with Bristol stool types 1 or 2 and >25% with Bristol stool types 6 or 7.
IBS Unspecified (IBS-U)	Patients who meet diagnostic criteria for IBS but whose bowel habits cannot be accurately classified into 1 of the 3 groups above.

^1^ Predominant bowel habits are based on stool form on days with at least one abnormal bowel movement. IBS subtypes can only be confidently established when patients are evaluated off all medications to treat bowel habit abnormalities.

**Table 2 microorganisms-10-02338-t002:** Lewis and Secretor status of mothers’ influences HMO composition of breastmilk ^1^.

Gene	Lewis Gene+	Lewis Gene−
Secretor gene+	Se + Le+ Secrete all HMOs	Se + Le– Secrete some HMOs (2′FL, 3′-FL, LNFP-I, LNFP-III)
Secretor gene−	Se–Le+ Secrete some HMOs (3′FL, LNFP-II and LNFP II)	Se–Le– Secrete some HMOs (3′FL, LNFP-III and LNFP-V)

^1^ Adapted from [94]. LNFP: Lacto-N-fucopentaose.

## Data Availability

Not applicable.

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
