# Peer review of "The Potential Role of Human Milk Oligosaccharides in Irritable Bowel Syndrome"

_microorganisms, 2022, doi:10.3390/microorganisms10122338_

Round 1
Reviewer 1 Report
The manuscript addresses an interesting topic and its content is quite interesting.
The use of probiotics like Human Milk Oligosaccharides, is discussed as a potential treatment/prevention of Irritable Bowel Syndrome, taking into account the data available in the literature.
The following suggestions are made for improving the manuscript:
- The manuscript is designed with a structure that is not very suitable for a review article. The Discussion must be "synergic" with the rest of the sections.
- Line 60: "NICE guidelines".Please use abbreviations after the first citation.
- Section 2.2 of the manuscript should be improved. Some recent articles may help to improve the manuscript.
- Section 3. I would suggest including a table with pharmacological and non-pharmacological treatments currently used for the management of IBS.
- Conclusions should be greatly improved.
- Future perspectives must be clearly exposed by the authors, thus improving the manuscript.
Author Response
Thank you for your valuable feedback. We have made some changes to the manuscript which will hopefully improve it.
The structure of the discussion has been reviewed to be more synergistic and flow with the rest of the manuscript.
Line 60: NICE guidelines abbreviation has been addressed.
Section 2.2 has been expanded to include more recent studies.
Section 3 now has a figure and is more comprehensive of non-pharmacological and pharmacological IBS treatment options.
Conclusions and future perspectives have been developed and improved.
Reviewer 2 Report
Irritable bowel syndrome, IBS, is as the authors adeptly point out a common health condition which costs great health care monies. The authors present a very comprehensive and well constructed consideration for the use of human milk oligosaccharides (HMO) as prebiotics for certain forms of IBS. The authors present a very good background of the types of IBS, their similarities, the incomplete understanding of pathogenesis, and abilities of certain treatments. The literature citation is good. Inclusion of Rome criteria with insightful analysis for treatment is also commended. The finding that HMO are of significant benefit in some IBS forms is cited and promising in certain studies.
One item deserving of greater expansion is why Bifidobacteria expand and why their increase may explain the benefit of HMO. Some of this incorporates the biochemistry of Bifidobacteria and this should be expanded. As in the first development of the biome of the neonatal human intestine, the ability of Bifidobacteria to metabolize oligosaccharides, particularly fructose oligosaccharides is unique. The metabolites generated are important to feed other bacteria including the beneficial Lactobacillus types.
Another minor point that may enhance the manuscript is why should human based oligosaccharides be used over plant based? Inulin is a higher fructose oligosaccharide that is known to be a good inducer of Bifidobacteria and promotes their therapeutic benefit. Natural sources of HMO are limited, bovine milk oligosaccharides are produced by the agricultural source and have also been shown to induce Bifidobacteria and then later Lactobacilli and have beneficial effects. Notably this literature is both for adults with a number of intestinal disorders, including IBS, but also for neonates and formula feeding.
Author Response
Thank you for your valuable feedback. We have made some changes to the manuscript which have improved it.
The biochemistry of bifidobacteria has been developed and available in vitro studies have been cited as evidence.
A point has been raised to explain why HMOs may be more suitable for IBS than inulin in terms of stimulating bifidobacteria, without aggravating IBS symptoms.